# Computational and Preclinical Analysis of 2-(4-Methyl)benzylidene-4,7-dimethyl Indan-1-one (IPX-18): A Novel Arylidene Indanone Small Molecule with Anti-Inflammatory Activity via NF-κB and Nrf2 Signaling

**DOI:** 10.3390/biomedicines11030716

**Published:** 2023-02-27

**Authors:** Reem M. Gahtani, Ahmad Shaikh, Hossam Kamli

**Affiliations:** Department of Clinical Laboratory Sciences, College of Applied Medical Sciences, King Khalid University, Abha 61413, Saudi Arabia

**Keywords:** arylidene, akt, NF-κB, Nrf2, RBL-2H3, basophils, mast cells, neutrophils, inflammation

## Abstract

**Simple Summary:**

Inflammatory responses are recorded in many dreadful diseases. As the presently used mainline anti-inflammatory treatments are proven to have adverse short- and/or long-term side effects, the search for alternative anti-inflammatory agents that may possess lesser side effects is on constant demand. This study evaluates IPX-18, a novel arylidene indanone small molecule, due to its anti-inflammatory activity mediated by NF-κB and Nrf2 signaling. These findings provide new insights for future research on this molecule for its development as a novel anti-inflammatory agent to treat several diseases.

**Abstract:**

**Background:** The adverse effects of anti-inflammatory drugs urges the search for new anti-inflammatory agents. This study aims at the preclinical analysis of the in-house synthesized small molecule IPX-18. Human whole blood (HWB), peripheral blood mononuclear cells (PBMCs), and neutrophils were used. Rat basophil cells (RBL-2H3) were used to assess degranulation. Binding stability to NF-κB-p50 was predicted using computational docking and molecular dynamic simulations. Essential signaling proteins were evaluated through flow cytometry. **Results:** IPX-18 inhibited the release of TNF-α with an IC_50_ value of 298.8 nM and 96.29 nM in the HWB and PBMCs, respectively. The compound depicted an IC_50_ value of 217.6 nM in the HWB and of 103.7 nM in the PBMCs for IFN-γ inhibition. IL-2 release and IL-8 release were inhibited by IPX-18 in the HWB and PBMCs. The compound controlled the migration of and the elastase in the activated neutrophils. The IC_50_ value for basophil activation through the FcεRI receptor assay was found to be 91.63 nM. IPX-18 inhibited RBL-2H3-degranulation with an IC_50_ value of 98.52 nM. The computational docking analysis predicted that IPX-18 would effectively bind NF-κB-p50. NF-κB-phosphorylation in the activated RBL-2H3 cells was decreased, and the levels of nuclear factor erythroid 2-related factor 2 (Nrf2) were increased with IPX-18 treatment. **Conclusions:** IPX-18 demonstrated efficacy in mediating the effector cells’ inflammatory responses through NF-κB/Nrf2 signaling.

## 1. Introduction

Inflammation is regarded as a cellular or biochemical response to any endogenous/exogenous agonists to maintain the body’s homeostasis [1,2]. Inflammatory responses are recorded in many diseases, including asthma, cancer, chronic inflammatory diseases, atherosclerosis, diabetes, autoimmune and degenerative diseases, etc., [3]. The adverse short- and/or long-term side effects of treatment regimens include cardiovascular and gastrointestinal complications, growth restrictions, insulin resistance, and neurodevelopmental disability [4]. Therefore, the search for alternative anti-inflammatory agents that may possess lesser side effects is on constant demand.

The nuclear factor kappa light-chain enhancer of activated B cells, or NF-κB, is a transcription regulator that is activated by various stimuli, including cytokines. Its stimulation with either IL-1β or TNFα activates the IκB kinase (IKK) complex, which then mediates the phosphorylation, ubiquitination, and degradation of the IκB molecule, which, in turn, releases NF-κB. NF-κB, when translocated to the nucleus, binds to the κB motifs in the promoter region to induce the transcription of target genes [5]. On the other hand, accumulating evidence favors the idea that Nrf2 (nuclear factor erythroid 2-related factor 2) plays a central role in anti-inflammatory functions. [6]. Studies indicate that Nrf2 negatively controls the NF-κB signaling pathway through more than one mechanism. Firstly, Nrf2 inhibits oxidative-stress-mediated NF-κB activation by decreasing the intracellular ROS levels [7]. Secondly, Nrf2 inhibits the proteasomal degradation of IκB-α to prevent the nuclear translocation of NF-κB [8]. An increase in the Nrf2 levels induces the cellular HO-1 levels and, thereby, phase II enzyme expression, which, in turn, blocks the degradation of IκB-α [9]. Alongside this, NF-κB decreases free CBP, which is a transcriptional coactivator of Nrf2, by competing with the CH1-KIX domain of CBP while also promoting the phosphorylation of p65 at Ser276, which, in turn, prevents CBP from binding to Nrf2 [10]. Accumulating evidence also suggests that Nrf2 counteracts the NF-κB-driven inflammatory response by competing with the transcription coactivator cAMP response element (CREB) binding protein (CBP) [11,12]. Another interesting study described that the Akt (PKB)-dependent inactivation of GSK3β (Glycogen synthase kinase 3 β) leads to an anti-inflammatory response by inactivating NF-κB and activating CREB and Nrf2 [13]. While it is presumed that the Nrf2 and NF-κB signaling pathways conjoin, therefore maintaining the physiological homeostasis of inflammatory regulation, targeting these signals have been proven worthy for controlling inflammation.

Recent advances in medicinal chemistry have contributed an array of small bioactive molecules to treat various diseases. Arylidene indanone is one such class of small molecules that is identified to have potent bioactivity against several diseases, including tuberculosis [14], degenerative diseases [15], hypoglycemia [16], lipidemia [17], and cancer [18]. 2-benzylidene-1-indanone derivatives were reported to be anti-inflammatory agents for the treatment of acute lung injury [19]. On the other hand, it is established that small-molecule anti-inflammatory drugs have lesser adverse effects with uncompromised efficacy [20]. Therefore, screening such small molecules for anti-inflammatory properties stands to be essential in searching for effective and economical novel anti-inflammatory compounds. The current study evaluates IPX-18 (Figure 1), one potentially active arylidene small molecule, due to its anti-inflammatory properties.

## 2. Materials and Methods

### 2.1. Materials

Reagents and chemicals were procured from Sigma–Aldrich (Burlington, MA, USA). RBL-2H3 cell line was obtained from ATCC (Manassas, VA, USA). ELISA kits were from eBioscience (San Diego, CA, USA). The Flow CAST^®^ kit was purchased form Buhlmann Diagnostics Corp (BDC) (Amherst, NH, USA). Migration inserts and 96-well plates were obtained from Nunc corp., Thermo Fischer Scientific (San Diego, CA, USA). Phospho-NF-κB p65 (S529) PE antibodies were from Thermo Scientific (San Diego, CA, USA). Recombinant PE anti-Nrf2 antibody was from Abcam (Fremont, CA, USA). Annexin V assay reagent was from Merck Millipore (Temecula, CA, USA).

### 2.2. Methods

#### 2.2.1. Synthesis of 2-(4-Methyl)benzylidene-4,7-dimethyl Indan-1-one

Schematic of synthesis of the small molecule is provided in Appendix A. Synthesis was initiated through the preparation of Baylis–Hillman adduct of p-tolualdehyde and t-butyl acrylate. This reaction was carried out on a solid-phase silica gel in presence of DABCO as catalyst. The reaction mixture was washed with ethyl acetate and was dried. After removing the solvent, crude product was chromatographed. Following this, the obtained hydroxy ester was dissolved in p-xylene and was treated with a catalytic amount of sulphuric acid under reflux. The solvent was removed under reduced pressure, and the residue was treated with TFAA in ethylene dichloride under reflux. Purification was carried out through column chromatography.

#### 2.2.2. Ethical Approval

This study was approved by the Research Ethics Committee, Deanship of Scientific Research, King Khalid University, Abha, Saudi Arabia (Reference No. ECM#2020-0913 (22 June 2020).

#### 2.2.3. Cell Culture

Eagle’s Minimum Essential Medium (supplemented with 10% FBS, 100 U/mL of penicillin, and 100 U/mL of streptomycin) was used for the growth of RBL-2H3 cells, and cell culture was maintained following the standard protocols.

#### 2.2.4. Annexin V Assay for Apoptosis/Cell Death

The assay was carried out by using Annexin V detection kit as per the manufacturer’s instructions. RBL-2H3 cells and PBMCs were treated with 100 nM or 250 nM IPX-18 along with suitable DMSO controls and incubation for 24 h. Cells were then treated with 0.25 μg/mL of Annexin V reagent for 15 min in the dark. After two washes in sterile PBS, cells were resuspended in kit buffer containing 0.5 μg/mL of propidium iodide. Ten thousand events were acquired on a Guava easyCyte™ flow cytometer. Data analysis was carried out with InCyte software (Version 6.0) to differentiate healthy and apoptotic cells (early and late apoptosis).

#### 2.2.5. Cytokine Assays

Cytokine production was performed using whole blood and PBMCs. Density gradient centrifugation was used to isolate PBMCs from whole blood as described elsewhere [21]. 0.5 × 10^6^ PMBCs/mL and HWB were resuspended in RPMI-8226 full growth media (1:3.5 dilution) and were seeded in 96-well tissue culture plates following incubation for 1 h at 37 °C with CO_2_. IPX-18, at the desired concentration, was applied to the wells in triplicate (*n* = 8) and was incubated for 30 min. Following this, PBMCs and HWB were induced with 50 ng/mL of phorbol-12-myristate-13-acetate (PMA) and 5 μg/mL of phytohaemagglutinin (PHA). All plates were incubated for 24 h for cytokine production, they were centrifuged at 4000 rpm for 10 min, and the supernatant was stored at −80 °C until further analysis. ELISA was performed per manufacturer’s instructions, and IC_50_ was calculated using Graphpad Prism (version 6.0).

#### 2.2.6. Isolation of Neutrophils from HWB

Neutrophil isolation from HWB was performed as described by previously established protocol [22]. Briefly, HWB, added to dextran sulphate, was kept at room temperature for 60 min. At 4 °C, the supernatant was centrifuged at 1150 rpm for 12 min. After adding 0.6 M KCl, it was again centrifuged at 1300 rpm with HiSep density gradient. Cells were collected in ice-cooled HBSS. A 2 × 10^5^ cells/mL density was adjusted for further neutrophil-based assays.

#### 2.2.7. Neutrophil Migration Inhibition Assay

To a 24-well plate, 249 μL of HBSS was added followed by 1 μL of 10 nM of N-formylmethionyl-leucyl-phenylalanine (fMLP). The neutrophil cell suspension was added to inserts and was placed in these wells. A total of 249 μL to 1 μL of DMSO, or several concentrations of IPX-18, was applied and was allowed to migrate for 1.5 h at 37 °C in a CO_2_ incubator. Inserts were stained with crystal violet solution and were eluted, and elution absorption was quantified spectrometrically at 410 nM.

#### 2.2.8. Neutrophil Elastase Assay

This assay was conducted as described elsewhere [23]. A total of 49 μL of neutrophils (2 × 10^5^ cells/mL) was added to 96-well plate and was treated with 1 μL of IPX-18 at different concentrations. A total of 200 μL of master mix that contained 10 μM fMLP, 200 μg/mL of CytochalasinB, 1 mM Sodium azide, and 1 mg/mL of L-Methionine was used to activate neutrophils for 1.5 h at 37 °C. After centrifugation at 1000 rpm for 5 min, 90 μL of supernatant was transferred to each well of a new 96-well plate. Following this, all wells were filled with 10 μL of 10 mM substrate and were incubated for 1 hour at 37 °C. The plate was read for absorbance at 410 nM.

#### 2.2.9. Basophil Activation Assay (BAT assay) in HWB

Flow CAST^®^ assay kit was used as per the manufacturer instructions. Human blood was incubated with IPX-18 for fifteen minutes and was treated with antihuman-FcεRI. After this, stimulation buffer and staining reagent were added and incubated in the dark for another fifteen minutes. Post centrifugation, the cells were suspended with 1 mL of PBS and were acquired using flow cytometer. The percentage of CD63 cells was analyzed.

#### 2.2.10. Cell Viability Assay

The cell viability was measured using MTT assay [24]. After preincubation for one hour with IPX-18, RBL-2H3 cells were activated for four hours with DNP-BSA. Furthermore, 100 μL of MTT (1 mg/mL) was added and incubated for 4.5 h. Formazan products were dissolved in DMSO and were read for absorbance at 560 nm.

#### 2.2.11. TNF-α and Degranulation in RBL-2H3 Cells

In RBL-2H3 cells, IgE-degranulation was analyzed by following the protocol stated previously by Naal et al. [25]. RBL-2H3 cells were seeded at a concentration of 2 × 10^5^ cells/mL in 24-well plates. These cells were sensitized with 1 μg/mL of IgE and were incubated overnight at 37 °C. Next day, the medium was aspirated and replaced with PIPES buffer and was further incubated at 37 °C for 15 min. After treating different concentrations of IPX-18 for one hour, anti-DNP-BSA was added for four hours. Supernatants from one set of experiments were stored at −80 °C and were used for TNF-α estimation through ELISA as per manufacturer protocol. Fresh supernatants from other set were removed from wells and were incubated with fluorescence substrate at 37 °C for 1 h. Fluorescence intensity was read at 450 nm (emission) and 360 nm (excitation).

#### 2.2.12. Structure Retrieval and Protein–Ligand Docking

The three-dimensional structure of p50 was retrieved from PDB databank (PDBID: 1SVC). Docking protein was prepared by adding polar hydrogens using Discovery Studio Visualizer. IPX structure was also prepared using Discovery Studio Visualizer. Docking was performed using SiBDOCK web server at www.sibiolead.com (accessed on 3 September 2020), which uses autodock vina docking protocol [26,27]. Docking grid box was set to focus on p50’s DNA-binding region, and box dimension was set to 20. Docked complex was analyzed using Discovery Studio Visualizer. Version. 19.1.0.18287. 

#### 2.2.13. Molecular Dynamic Simulation

Simulation was performed using an automated protein–multiligand protocol developed at SiBioLead www.sibiolead.com (accessed on 3 September 2020) that uses GROMACS simulation package. Protein–ligand complex was immersed in a triclinic box containing SPC water molecules and NaCl as counterions. Furthermore, 0.15 M NaCl was added to the simulation system to maintain physiological conditions. OPLS/AA forcefield was applied to the simulation system. Before simulation, system was equilibrated for 300 ps using NVT/NPT protocol. Simulation was carried out using leapfrog integrator in a GPU-based Linux environment. Simulation trajectories were analyzed using GROMACS in-built results analyses package.

#### 2.2.14. Flow Cytometry

Essential cellular protein singling was detected through flow cytometry. Initially, RBL-2H3 cells were sensitized with DNP-IgE, and then they were treated with doses of IPX-18 for 30 min and were induced with DNP-BSA for 4 h. Furthermore, cells were stained for 15 min with NF-κB p65 (S529) PE antibody or anti Nrf2 PE antibody in buffer reagent (Thermo scientific, San Diego, CA, USA). The cells were washed twice to remove excess stain and were resuspended in HBSS buffer. 5000 events were acquired with Guava EasyCyte™ flow cytometer, Merk Millipore (Temecula, CA, USA). Positively gated cells were further read for NF-κB/p65s529 or Nrf2 using InCyte software from Merk Millipore (Temecula, CA, USA).

#### 2.2.15. Statistical Analysis

The experiments were performed in triplicate, and all data were represented as mean ± S.D. Graphpad Prism 6.0 (La Jolla, CA, USA) was used for statistical examinations. To compare the differences between the two groups, Student’s *t*-test was used for this analysis. ANOVA was used for determining the difference between three or more variants. Statistical significance was considered to be *p* < 0.05 (*).

## 3. Results

### 3.1. Chemistry of Synthesized Small Molecule

A 1H NMR of the synthesized compound is attached in Appendix A. The product showed IR absorptions at 1642, 1712, and 3414 cm^−1^, indicating this to be an α,β-unsaturated hydroxy ester. The compound was t-butyl-3-hydroxy-2-methylene-3-(4-methyl-phenyl) propanoate. This was confirmed with a PMR spectrum. A multiplet around 7.15–7.35 δ for four protons indicated the phenyl group. A singlet for one proton at 6.23 δ indicated the lone benzylidene proton, which was allylic, as well as the hydroxy methyl proton. Each of the two vinylic protons appeared as a singlet at 5.71 and 5.47 δ. A singlet at 2.4 δ indicated the three Ar–CH_3_ protons, and a singlet for nine protons indicated the t-butyl group, which appeared at 1.6 δ. The IR spectrum showed absorptions at 1689.5 cm^−1^ and 1624 cm^−1^, indicating this to be an aromatic α,β-unsaturated ketone. The PMR spectrum showed a multiplet around 7.7–7.53 δ for seven protons. These were the six aromatic protons and the one vinylic proton. A two-proton singlet at 3.75 δ indicated the benzylic and allylic methylene protons. The Ar–CH_3_ appeared as a three-proton singlet at 2.67 δ. Two singlets appeared at 2.37 and 2.32 δ for six Ar–CH_3_ protons. A signal at 195.74 δ in the ^13^C NMR spectrum indicated the α,β-unsaturated carbonyl carbon. The presence of the Ar–CH_3_ carbons appeared at 17.7, 18.31, and 21.59 δ. The aromatic carbon atoms appeared around 129 to 149.16 δ. The m/e value of the compound corresponded to the molecular weight of 262.5, and the elemental analysis agreed with the molecular formula of the compound. The following values were calculated: C, 86.92%; H, 6.91%. The following values were found: C, 86.98%; H, 6.94%. Based on the above data, the compound was identified to be 2-(4-methyl) benzylidene-4,7-methyl indan-1-one.

### 3.2. Nontoxicity of IPX-18

Prior to testing IPX-18’s anti-inflammatory effects, the nontoxicity of the compound was assessed through the annexin V assay. For this, IPX-18, at different concentrations, was incubated with the RBL-2H3 cells and the human primary leucocytic cells (PBMCs) with untreated controls for 24 h. The compound did not induce any early- or late-phase apoptosis in these cells when compared to the untreated controls (Appendix A).

### 3.3. IPX-18 Attenuated Proinflammatory Cytokine Responses in Human Whole Blood and Peripheral Nucleocytes

The HWB and PBMCs were used in the initial screening of IPX-18 for proinflammatory cytokine inhibitions. The compound inhibited the release of TNF-α with an IC_50_ value of 298.8 nM and 96.29 nM in the HWB and PBMCs, respectively (Figure 2a). IPX-18 inhibited IFN-γ with an IC_50_ value of 217.6 nM in the HWB and of 103.7 nM in the PBMCs (Figure 2b). Inhibition of IL-2 and IL-8 in the HWB by IPX-18 was evident with IC_50_ values of 416.0 nM and 336.6 nM, respectively (Figure 2c,d). The IC_50_ values for these cytokines in the PBMCs were 122.9 nM for IL-2 and 105.2 nM for IL-8 (Figure 2c,d).

### 3.4. Effect of IPX-18 Treatment on Activated Neutrophils

The results of elastase activity and neutrophil migration confirmed the dose-responsive efficacy of the compound in inhibiting neutrophil migration (Figure 3a). Similarly, IPX-18 inhibited the elastase exocytosis of the stimulated neutrophils in a dose-responsive way (Figure 3b).

### 3.5. IPX-18 Effectively Inhibited the Activation of Basophils

The basophils in the HWB were stimulated by crosslinking the IgE binding receptors and anti-FcεRI monoclonal antibodies. When provoked, the CD63 surface receptors that were expressed in these cells were analyzed through flow cytometry. CCR3 positive cells were gated to determine CD63 + ve populations. The results demonstrated a loss in the CD63 cell population (Figure 4) with IPX-18 treatment. When assessed for dose dependency, the IC_50_ value for IPX-18 was found to be 91.63 nM (Figure 4).

### 3.6. Dose Tolerance of IPX-18 on Normal and Stimulated RBL-2H3 Cells

Prior to analyzing the effects of IPX-18 in the RBL-2H3 cells, an MTT assay was used to evaluate the viability of the RBL-2H3 cells. The viability of the RBL-2H3 cells was unaltered with up to 250 nM of IPX-18 (Figure 5a) for 24 h. To check out if the stimulation could bring out any toxic effect in these cells, different doses of IPX-18 were assessed for cytotoxicity with 0.025 μg/mL of DNP-BSA (dinitrophenyl human and bovine serum albumin) for 4 h in 1 μg/mL of the anti-DNP IgE antibody (antidinitrophenyl anti-IgE antibody) pretreated RBL-2H3 cells. Results indicated no loss of cell viability with up to 250 nM of IPX-18 in the presence of the stimulator (Figure 5b).

### 3.7. IPX-18 Inhibited TNF-α Release and Degranulation in Anti-DNP/IgE of Sensitized RBL-2H3 Cells

A reduction in TNF-α release (Figure 5c) in the cell supernatant was evident with IPX-18 treatment. In order to check the effect of the compound in the degranulation process, we assessed the release of β-glucuronidase, an essential degranulation enzyme, in the RBL-2H3 cells after stimulation with DNP-BSA. IPX-18, with an IC_50_ value of 98.52 nM, could dose-dependently control RBL-2H3 degranulation (Figure 5d).

### 3.8. Protein–Ligand Docking of IPX-18 to p50 Subunit of NFkB

In order to understand the mode of action, we performed the computational modeling and docking of IPX-18 to the NFkB p50 subunit. For this purpose, we used an experimental structure of p50 bound to DNA from the PDB databank (PDBID: 1SVC). An analysis of the retrieved p50-DNA complex suggested that targeting the amino acid residues that participate in DNA interactions would repress p50 DNA binding. We previously showed that targeting the DNA-binding region of p50 significantly induced apoptosis [1]; therefore, in this work, we targeted the same region for our docking studies (Figure 6a). An analysis of the DNA-binding interface of p50 identified crucial residues and a pocket that could be targeted with a small molecule (Figure 6b). We performed the protein–ligand docking of the IPX-18 molecule to the target site in the p50 protein. The results indicated that IPX-18 fits well in the DNA-binding interface of the p50 protein (Figure 6c). The docking score (binding energy) of IPX-18 to p50 was predicted to be −6.2 kcal/mol, which was comparable and better than the standard compound used in our previous work [1]. A protein–ligand interaction analysis of the p50 bound IPX-18 complex indicated that IPX-18 interacts with the crucial residues involved in DNA binding (Figure 6d).

### 3.9. Molecular Dynamic Simulation Predicted IPX-18 Binding Stability

In order to assess the binding stability of the IPX-18 complex with the p50 subunit of NFkB, we performed a 100 ns atomistic molecular dynamic simulation of the IPX-18 NFkB-p50 complex using the GROMACS simulation package. A simulation trajectory analysis at different timepoints predicted that IPX-18 would bind stably to the predicted binding site of the p50 subunit. Comparing the 50 ns and 100 ns frames with 0 ns, (i.e., the complex before the simulation), the IPX-18 molecule rotated from its initial docked position; however, the residue interactions of IPX-18 remained the same, indicating that IPX-18 attained a favorable conformation during the simulation (Figure 7a–c). The simulation trajectory video shows the binding stability of IPX frame by frame (Appendix A).

### 3.10. Efficacy of IPX-18 on Key Signaling Proteins of the Inflammatory Pathway

The phosphorylation of NF-κB in the stimulated RBL-2H3 cells was estimated through flow cytometry. DNP-BSA stimulated the RBL-2H3 cells and induced the phosphorylation of NF-κB at 4 h postinduction (Figure 8a). Preincubation with IPX-18 for 30 min before the stimulation reduced the phosphorylation of NF-κB (Figure 8a). Similarly, the expression levels of the nuclear Nrf2 proteins were upregulated by IPX-18 (Figure 8b) with the DNP-BSA stimulation of the RBL-2H3 cells sensitized previously with IgE.

## 4. Discussion

Chronic inflammation leads to severe conditions that may end up with complex manifestations [28]. Though there are several anti-inflammatory drugs available in the market, their long-term administration may cause several unfavorable side effects, including organ toxicities, ulcerations, and bleeding [28]. Therefore, the search for novel anti-inflammatory compounds is on demand. This research focuses on the preclinical evaluation of a novel arylidene indanone moiety as a potential agent against inflammatory responses. Cytokines constitute the prime physiological defense against any stimuli during the innate and adaptive inflammatory responses [29]. It is reported that inflammation is a complex phenomenon that involves the interplay between neutrophils, basophils, mast cells, etc., which is commonly orchestrated via cytokines [30]. Therefore, any agent that can bring down the levels of the cytokines in the activated whole blood or PBMCs can be regarded as having control over the inflammatory process. TNF-α is responsible for increasing the number of effector cells at the inflammation site [31], while IFN-γ and IL-2 are proven to drive monocytes, macrophages, and lymphocytes during autoimmune-related inflammation [32]. On the other hand, IL-8 invades the neutrophils and macrophages at the inflammatory site and further drives the process [33]. The efficacy of IPX-18 in deteriorating these cytokine levels could thereby be related to the molecule’s anti-inflammatory potency.

During acute inflammation, polymorphonuclear neutrophils constitute the first line of defense [34]. As observed in the neutrophil migration assay, the influx of the activated neutrophils was attenuated by IPX-18. Research indicates that TNF-α is a chief proinflammatory cytokine which plays a role in leukocyte rolling and adhesion [35]. Furthermore, studies have shown that the inhibition of NF-κB by impeding p50 nuclear translocation via interleukin-10 exerts anti-inflammatory effects [36]. The present study observed neutrophil attenuation and the migration inhibition of TNF-α by IPX-18. Neutrophil elastase is an essential factor for the progression of many types of inflammation and is well liked with respect to neutrophil migration and activation [2]. The dose-dependent elastase inhibition of IPX-18 observed in this study could be attributed to the attenuation of both the compound and the neutrophil infiltration functions. Similarly, basophil activation is yet another important mechanism in inflammatory responses [37]. The lysosomal membrane protein CD63 was well expressed on the activated basophil surface [38], which accounted a direct measure of the activated basophils using flow cytometry [39]. In order to discriminate the CD63 positive basophils from the other cell types, such as eosinophils, T cells, and neutrophils, CCR3 was used as a gate marker in the flow cytometry assay [40]. Consequently, the CCR3 + CD63 positive population accounted a direct measure for the activated basophils expressing the high-affinity IgE receptor FcεRI [40,41]. Therefore, several IgE mediated inflammatory responses were diagnosed using the FcεRI-BAT assay [42]. The observations in the present study stand well with reports indicating the efficacy of IPX-18 in inhibiting basophils’ activation.

Mainstream allergy diseases are mediated by immunoglobulin E (IgE)-dependent reactions [43]. These allergic inflammations are provoked in dual phases, the initial induction phase followed by the effector phase [44]. During the initial phase, IgE couples with the surface FcεRI-IgE receptors of mast cells and basophils [44]. During the second phase, the adjacent IgEs of different presensitized basophils/mast cells are crosslinked by the allergen to release proinflammatory cytokines [44]. At the end of the effector phase, secondary cells such as the neutrophils are recruited and are activated by the effector substances released by the activated mast cells and basophils [45]. Together, basophil stimulation and mast cell stimulation in addition to the downstream activation of neutrophils are centrally driven by IgE orchestrated mechanisms. Hence, this study evaluated the efficacy of IPX-18 in activated rat basophils (RBL-2H3 cells) on IgE mediated stimuli. The observations indicated the nontoxicity of IPX-18 in the normal RBL-2H3 cells and activated RBL-2H3 cells, thereby indicating its therapeutic safety. Additionally, the compound also dose-dependently inhibited TNF-α release and degranulation in the activated RBL-2H3 cells, proving the anti-inflammatory efficacy of IPX-18 in IgE driven inflammatory manifestations.

NF-κB is a multifunctional protein located downstream of Akt and is considered an ideal proinflammatory marker [46]. Studies show that NF-κB is activated by heterodimer formation between p50 and p65 [47]. Therefore, the p50 subunit of NF-κB could be targeted to impede complex formation, thereby deactivating NF-κB activation. Our computational analysis predicted that IPX-18 would effectively bind NF-κB-p50. The in vitro results of this study agreed with the computational prediction, where the inhibition of NF-κB phosphorylation was evident with IPX-18 treatment. It has been previously shown that NF-kB/p65 antagonizes Nrf2-ARE signaling [47]. Nrf2 activity is critical for the body’s defense against oxidants, carcinogens, and inflammatory insults, and the overexpression of p65 suppresses Nrf2 levels/activity. Furthermore, it is reported that Nrf2 signaling reversely affects the regulation of the inflammatory responses arbitrated by NF-κB [47]. Thus, inhibiting p65 activity upregulates the Nrf2 levels. The observations of the present study stand well with the literature, where IPX-18 induced the elevation of the Nrf2 levels by translocating it into the activated RBL-2H3 cells’ nuclei. Therefore, it can be postulated that the efficacy of IPX-18 could be driven by the NF-κB pathway, causing the upregulation of Nrf2 proteins. 

## 5. Conclusions

In summary, IPX-18 demonstrated excellent efficacy in controlling various mediators of the inflammatory responses through the dephosphorylation of NF-κB and the upregulation of Nrf2 proteins. These findings provide new insights for future research on the molecule for its development into a novel anti-inflammatory agent to treat several diseases.

## Figures and Tables

**Figure 1 biomedicines-11-00716-f001:**
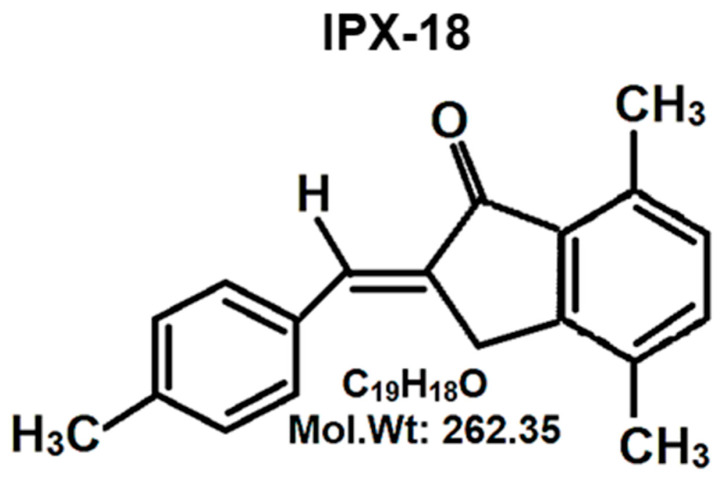
Chemical structure of IPX-18.

**Figure 2 biomedicines-11-00716-f002:**
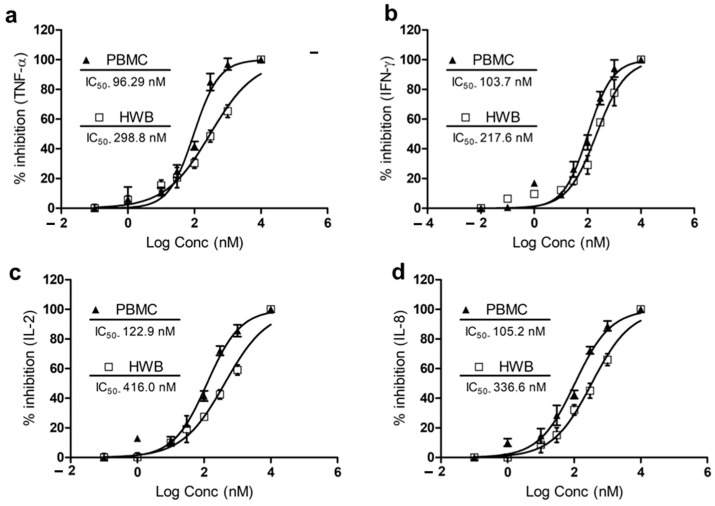
IC_50_ indices of IPX-18 for inhibition of (**a**) TNF, (**b**) IFN, (**c**) IL-2, and (**d**) IL-8 release in HWB (*n* = 8) and PBMCs (*n* = 8). Cytokine secretion was determined with ELISA kits. Results were expressed as the mean ± SD from eight donors performed in triplicates, and IC_50_ values determined by GraphPad prism software.

**Figure 3 biomedicines-11-00716-f003:**
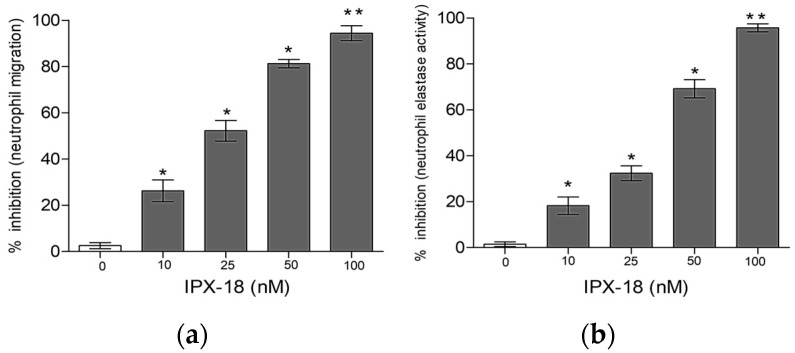
IPX-18 inhibited neutrophil (**a**) migration and (**b**) elastase exocytosis in a dose-dependent way. Results are expressed as mean ± SD from five donors. * *p* < 0.05 and ** *p* < 0.001 are considered statistically significant.

**Figure 4 biomedicines-11-00716-f004:**
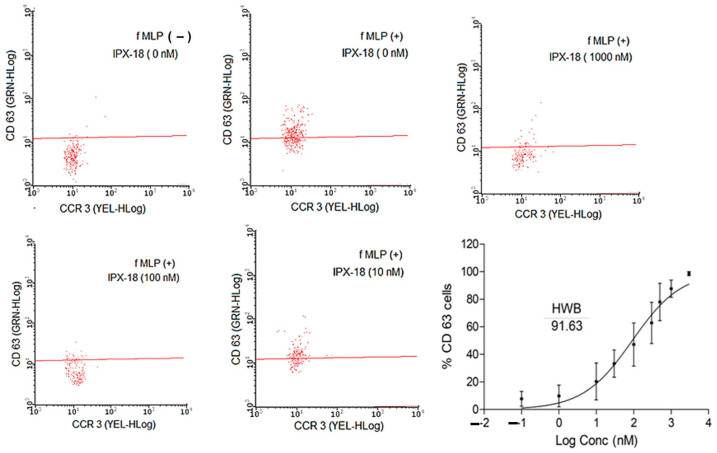
Inhibitory effects of different concentrations of IPX-18 on activated basophils. Stimulated by anti-FcεR1, a highly specific monoclonal antibody bound to the high-affinity IgE binding receptor and mimicked the bridging of the receptor caused by the allergen binding to two bound immunoglobulin E molecules. Representative histograms from basophil activation test (BAT) in HWB from five different donors are shown. Basophil activation was evaluated in CD63 expressed positive cells gated from CCR 3 positive population. IC50 of IPX-18 from five donors analyzed for CD63 positive cells with GraphPad Prism (6.0) software.

**Figure 5 biomedicines-11-00716-f005:**
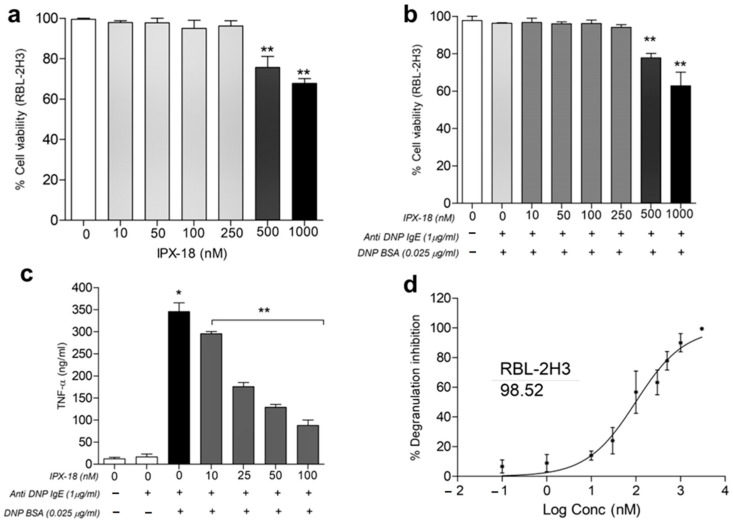
Cytotoxicity indices for IPX-18 at different concentrations in (**a**) normal RBL-2H3 cells and (**b**) anti-DNP IgE sensitized RBL-2H3 cells with DNP-BSA induction. Cell viability was measured through MTT assay, and results are expressed as mean ± SD (*n* = 3). Results are statistically significant at ** *p* ≤ 0.01. (**c**) Dose-dependent cytokine/degranulation inhibitory profile of IPX-18 in anti-DNP IgE sensitized RBL-2H3 cells stimulated by DNP-BSA. RBL-2H3 cells were sensitized overnight, were preincubated with different concentrations of IPX-18 for 1 h, and were stimulated with DNP-BSA for 4 h. (**c**) TNF-α was measured in the supernatants. Results are expressed as mean ± SD (*n* = 3), and statistical significance is at * *p* ≤ 0.05/ ** *p* ≤ 0.01. (**d**) RBL-2H3 cell degranulation in supernatant was assessed fluorometrically as measure of β-glucuronidase release using 4-methylumbelliferyl-beta-D-glucuronide substrate. Results are expressed as the mean ± SD, and IC_50_ values were determined with GraphPad prism (6.0) software.

**Figure 6 biomedicines-11-00716-f006:**
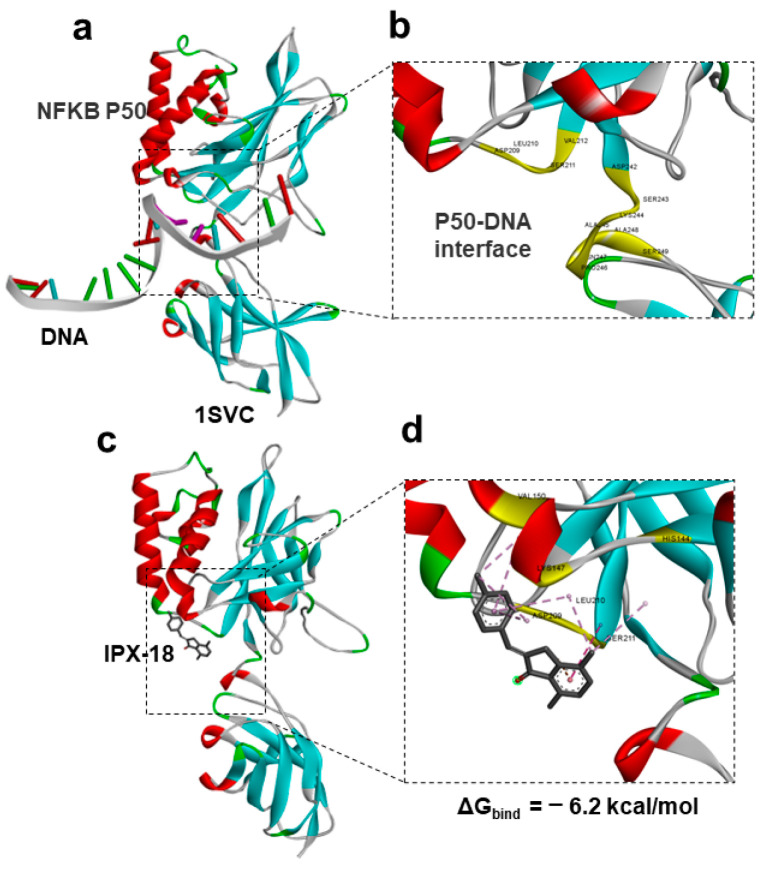
Computational docking of IPX-18 to NF-κB-p50. (**a**) Experimental structure of p50 bound to DNA from PDB databank (PDBID: 1SVC). (**b**) Identification of crucial residues and pocket for binding IPX-18. (**c**) IPX-18 fit well in the DNA-binding interface of p50 protein. (**d**) Protein–ligand interaction analysis of p50 bound IPX-18 complex.

**Figure 7 biomedicines-11-00716-f007:**
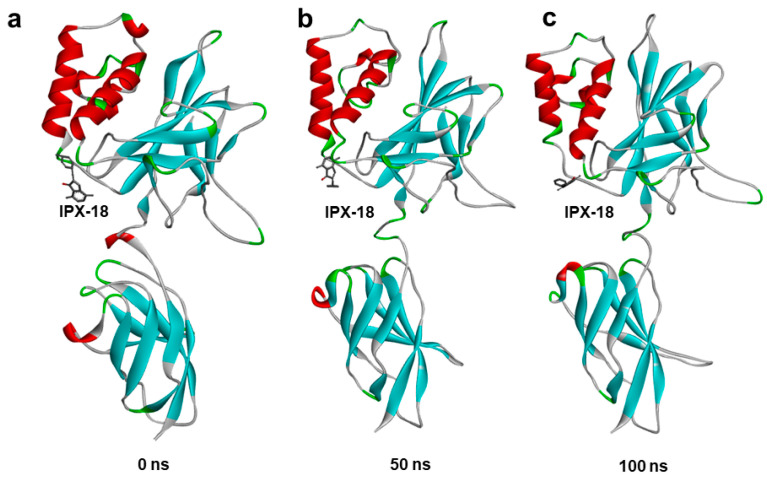
Molecular dynamics simulation of IPX-18 complex with NF-κB-p50: Ligand to protein root mean square deviation (RMSD) for 100 ns trajectories. Snapshots taken at different time points of (**a**) 0 ns, (**b**) 50 ns, and (**c**) 100 ns with simulation showing binding modes of IPX-18 to NF-κB-p50.

**Figure 8 biomedicines-11-00716-f008:**
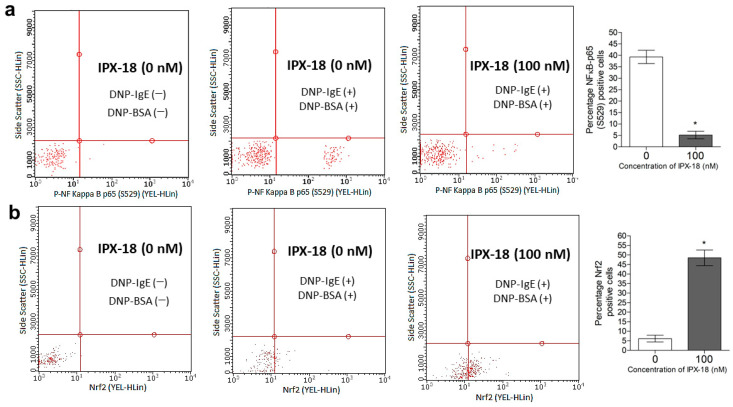
IPX-18 inhibited (**a**) p-NF κB [p65] and upregulated (**b**) Nrf2 in RBL-2H3 cells. RBL-2H3 cells were presensitized overnight with 1 μg/mL of anti-DNP IgE and were incubated with different concentrations of IPX-18 for 30 min followed by induction with 0.025 μg/mL of DNP-BSA for 4 h. Percentage of positive P-NFkB and Nrf2 cells was determined through flow cytometry. Representative graphs are presented. Numerical values in histograms are from three individual experiments. Statistical significance is at * *p* < 0.05.

## Data Availability

All data were used in this manuscript. The data that supports the findings of this study are available from the corresponding author upon reasonable request.

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
