# Peer review of "Computational and Preclinical Analysis of 2-(4-Methyl)benzylidene-4,7-dimethyl Indan-1-one (IPX-18): A Novel Arylidene Indanone Small Molecule with Anti-Inflammatory Activity via NF-κB and Nrf2 Signaling"

_biomedicines, 2023, doi:10.3390/biomedicines11030716_

Round 1

Reviewer 1 Report

In this manuscript, Gahtani et al provided experimental data showing the anti-inflammatory effect of a novel small molecule, IPX-18, which could inhibit the activation of a panel of effector leukocytes measure by cytokine production, migration and degranulation. The mechanism was predicted to be mediated by NF-κB and Nrf2 signaling, and experimental validation assays were designed. Despite the fact that this is an interesting study and we do need improved anti-inflammatory drugs with robust efficacy and a better safety profile, a number of technical issues has been spotted and needs to be addressed.

 Major concerns:

1.     A major confounding issue in the study is the possible effect of IPX-18 on the induction of cell death. Although the authors measured viability of RBL-2H3 cells following treatment with the drug, this does not necessarily support the non-viability compromising effect of IPX-18 on human cells. First of all, MTT is not sensitive enough to monitor early phases of cell death. MTT can overestimate the viability of cells because during the early stages of apoptosis the cell is still able to metabolise the tetrazolium salt (PMID 27774532). Proper early cell apoptosis assays should be performed for all the cell types analyzed. Secondly, treatment duration of RBL-2H3 with IPX-18 was much shorter than for the human cell activation assays, let alone the differences between the rat cell line and primary human cells. Usually, transformed cell lines resist cell death induction more robustly than primary cells. Therefore, it is not possible to extrapolate the viability result of RBL-2H3 to human leukocytes. Thirdly, in all the human cell studies, it seems that the IC50 values were similar for the same cells irrespective of the cytokine release measured, which indicates the involvement of a possible confounding element (e.g., cell death) in the analysis.

 2.     In various assays testing the effect of IPX-18, the incubation duration is different, ranging from 15 min to 1.5 h. What is the rationale for using the different incubation times? Has the IPX-18 been washed away after the incubation, before the next incubation stage for cell activation?

 Minor concerns

1.     Line 124, page 4: What does “anti-IgE-FceRI mean? For basophil activation, one should either use an antibody against human IgE or FceRI (recognizing an epitope that is not sterically blocked by bound IgE).

2.     There are many typos and grammatical errors in the text.

Author Response

In this manuscript, Gahtani et al provided experimental data showing the anti-inflammatory effect of a novel small molecule, IPX-18, which could inhibit the activation of a panel of effector leukocytes measured by cytokine production, migration and degranulation. The mechanism was predicted to be mediated by NF-κB and Nrf2 signaling, and experimental validation assays were designed. Despite the fact that this is an interesting study and we do need improved anti-inflammatory drugs with robust efficacy and a better safety profile, a number of technical issues has been spotted and need to be addressed.

We thank the reviewer for the input. We have addressed the concerns of reviewer by suitable revising the manuscript. Point-wise responses to the comments are provided.

 Major concerns:

  1. A major confounding issue in the study is the possible effect of IPX-18 on the induction of cell death. Although the authors measured the viability of RBL-2H3 cells following treatment with the drug, this does not necessarily support the non-viability compromising effect of IPX-18 on human cells. First of all, MTT is not sensitive enough to monitor early phases of cell death. MTT can overestimate the viability of cells because during the early stages of apoptosis the cell is still able to metabolize the tetrazolium salt (PMID 27774532). Proper early cell apoptosis assays should be performed for all the cell types analyzed.

We appreciate the reviewer for bringing out this point to assess the viability of cells. In the current study, RBL2H3 cells were used to assess degranulation and TNF alpha cytokine production under provoked conditions. We have shown the viability of RBL2H3 cells tolerating the doses of IPX-18 both individually and along with the inducer to rule out the possibility if the inducer had any effect on the cell viability at the actual treatment concentrations of further experiments. From the data of the MTT assay, IPX-18 individually or along with the inducer did not show a significant reduction in the cell viability up to 250 nM. Therefore, the cytokine estimation and degranulation experiments used 0 to 100 nM concentration of IPX-18, where a dose responsive inhibition of both cytokine production and degranulation were observed (5 c and d). In this regard, the reviewer has raised a point if the MTT assay is sensitive enough to detect the early apoptosis in RBL 2H3 cells and in case, if early apoptosis could be a reason for the observed inhibitory effects in both these assays. To confirm this, we have performed the Annexin V assay in RBL-2H3 cells and have included it in the revised manuscript as supplementary figure-3. From our observation, we did not observe a significant difference in the early/late phase apoptosis of RBL-2H3 cells when treated with IPX-18 with 100 nM or 250 nM concentrations, when incubated for 24h. Whereas the concentrations of IPX-18 in determining the degranulation and cytokines were lesser than 6 hours’ time.

  1. Secondly, treatment duration of RBL-2H3 with IPX-18 was much shorter than for the human cell activation assays, let alone the differences between the rat cell line and primary human cells. Usually, transformed cell lines resist cell death induction more robustly than primary cells. Therefore, it is not possible to extrapolate the viability result of RBL-2H3 to human leukocytes.

Similar to the first part of the query, we incubated 100 and 250 nM of IPX-18 with PBMCs (Primary cells) and have conducted Annexin V assay at the end of 24 h incubation (Supplementary figure 3). There was no significant difference in early or late phase apoptotic cell populations when compared to the IPX-18 untreated control cells.

  1. Thirdly, in all the human cell studies, it seems that the IC50 values were similar for the same cells irrespective of the cytokine release measured, which indicates the involvement of a possible confounding element (e.g., cell death) in the analysis.

With the observations of Annexin V assays in RBL 2H3 cells and primary human leucocytes, it can be inferred that the observed efficacy was not due to the cell death caused by IPX-18.

 4     In various assays testing the effect of IPX-18, the incubation duration is different, ranging from 15 min to 1.5 h. What is the rationale for using the different incubation times? Has the IPX-18 been washed away after the incubation, before the next incubation stage for cell activation?

We have used these incubation time based on our laboratory standardizations as previously established protocols. The drug incubation time was also based on recommendations of manufacturer where kits have been used. Washing of IPX-18 prior to activation was performed based on the assay requirements.  

 Minor concerns

  1. Line 124, page 4: What does “anti-IgE-FceRI mean? For basophil activation, one should either use an antibody against human IgE or FceRI (recognizing an epitope that is not sterically blocked by bound IgE).

We accept the comment and have now corrected it in revised manuscript.

  1. There are many typos and grammatical errors in the text.

We have revised the manuscript for grammar, typos and English language as suggested.

Reviewer 2 Report

In their work, the authors examined the cytotoxicity and anti-inflammatory effect of a newly synthesized compound, IPX-18, in various cell lines. They also performed a computational analysis of its potential impact on the NF-kB p50 DNA binding. 

After introducing some crucial changes, the work may be suitable for publication in Biomedicines.

1. Abstract/Introduction

NSAIDs (like naproxen and ibuprofen) and, after that, COXIBs are invariably the main groups of medications used to reduce inflammation processes. Corticosteroids are steroid hormones only when prescribed by a doctor in certain doses and only for certain conditions like, for example, RA, IBD, asthma, and allergies reduce inflammation but also suppress the immune system. It is a major mistake to claim that corticosteroids are the main-line anti-inflammatory drugs.

2. Introduction

It needs to be written better and indefinitely and sufficiently justify the research presented. It requires a description of the relationship between the inflammatory process and the NF-kB, Nrf2 signaling pathways, genes staying under its control, the connection that justifies examining both, and the level of cytokines described in the results. Do compounds from the group of arylidene indanones already have a proven anti-inflammatory effect? There also needs to be an explanation of why these particular cell lines were selected for the study.

'Dreadful diseases' is not a very professional term and needs rephrasing. Brain damage is not a disease; what does it refer to? By neuro diseases, did the authors mean neurological diseases, such as AD or VAD?

In line 53, a sentence begins that is not related to the diseases mentioned earlier.

3. Methodology

It has a lot of grammatical and punctuation errors.

Methods should be described in more detail in some subsections, e.g., 2.2.7. what kind of 'small changes' were introduced?

4. Results

The results describe the effect of the newly synthesized compound on inflammation markers in several cell lines. The results of the MTT assay should be placed on one figure for all tested cell lines, and the IC50 values preferably presented in a table to clearly show the differences in the cytotoxicity on different cell types.

Are the results presented in figure 2 from a total of 24 results (eight donors in triplicates)?

In figure 3 (panels a and b), are the authors sure that the differences were not statistically significant after applying IPX-18 at 10, 25, and 50 nm concentrations?

In Figure 4, the panels, their description, and an explanation of what f MLP means are missing.

In subsection 3.5, what does it mean that the cells tolerated the treatment up to 250 nm? The concentration of the compound assuring cell viability >70% justifies the use of such a concentration in the in vitro studies.

In subsection 3.7. what does the p50 bound mean? How is this binding region related to the described by authors apoptosis and inflammation? Does the 'small molecule' term always refer to the IPX-18 compound? It should be clear that the authors describe this compound. The description should also highlight how binding IPX-18 to the p50 subunit could theoretically affect inflammatory processes and why this particular subunit was studied and not p65.

In subsection 3.9. the authors present the phosphorylation of the p65 subunit. This subunit of NF-kB dimer is being phosphorylated directly in IKK-independent (atypical) signaling (after UV light, for example), while canonical NF-kB signaling is TNF alpha-mediated. The authors should also note whether and how Nrf2 activation is related to NF-kB and whether Nrf2 activation is beneficial in cancer cells. Are Nrf2 results from whole cell extracts? Description of Figure 3.9. suggests a statistical significance that is not indicated in the figure.

In general, all figures' captions should be more detailed.

What justifies the use of different concentrations of the compound in various presented methods?

Author Response

In their work, the authors examined the cytotoxicity and anti-inflammatory effect of a newly synthesized compound, IPX-18, in various cell lines. They also performed a computational analysis of its potential impact on the NF-kB p50 DNA binding. 

After introducing some crucial changes, the work may be suitable for publication in Biomedicines.

We thank the reviewer for the valuable comments and appreciation, and recommendation of our work. We have revised the manuscript as per the suggestions and have provided point wise responses to the comments as follows.

  1. Abstract/Introduction

NSAIDs (like naproxen and ibuprofen) and, after that, COXIBs are invariably the main groups of medications used to reduce inflammation processes. Corticosteroids are steroid hormones only when prescribed by a doctor in certain doses and only for certain conditions like, for example, RA, IBD, asthma, and allergies reduce inflammation but also suppress the immune system. It is a major mistake to claim that corticosteroids are the main-line anti-inflammatory drugs.

We accept the comment and have removed the claim in the revised manuscript

  1. Introduction

It needs to be written better and indefinitely and sufficiently justify the research presented. It requires a description of the relationship between the inflammatory process and the NF-kB, Nrf2 signaling pathways, genes staying under its control, the connection that justifies examining both, and the level of cytokines described in the results. Do compounds from the group of arylidene indanones already have a proven anti-inflammatory effect? There also needs to be an explanation of why these particular cell lines were selected for the study.

Thanks for providing insight to improve the manuscript. As suggested, we have improved the introduction part with new references. ( Page 2, Line 51 to 61 in revised manuscript).

'Dreadful diseases' is not a very professional term and needs rephrasing. Brain damage is not a disease; what does it refer to? By neuro diseases, did the authors mean neurological diseases, such as AD or VAD?

This part of introduction is modified as suggested by reviewer.

  1. Methodology

It has a lot of grammatical and punctuation errors. Methods should be described in more detail in some subsections, e.g., 2.2.7. what kind of 'small changes' were introduced?

We accept the comment and have modified accordingly in the revised manuscript. We have removed the “small changes” and have explained the method.  

  1. Results

The results describe the effect of the newly synthesized compound on inflammation markers in several cell lines. The results of the MTT assay should be placed on one figure for all tested cell lines, and the IC50 values preferably presented in a table to clearly show the differences in the cytotoxicity on different cell types.

We have tested the proinflammatory cytokine release effect of IPX-18 in human whole blood and PBMCs. The effect of the compound on neutrophil action like elastase release and migration was assessed. Human whole blood was used in the basophil activation test. All of these assays did not use cell lines. Cytokine release, degranulation process and protein signaling by flow cytometry were the assays performed on the RBL-2H3 cell line. Prior to this, MTT assay was performed in this cell line (Fig 5 and b) to check the cell viability with different doses of IPX-18 in the presence and absence of inducers. Therefore, we would like to explain that no IC50 in MTT assay was performed on different cell lines in this study.   

Are the results presented in figure 2 from a total of 24 results (eight donors in triplicates)?

As described in the method section, the results are the average of 8 donors ( Each seeded in triplicates of 96 well plate). As suggested, we have elaborated this in the revised manuscript.

In figure 3 (panels a and b), are the authors sure that the differences were not statistically significant after applying IPX-18 at 10, 25, and 50 nm concentrations?

We thank the reviewer for identifying this. All these were statistically significant. It is now corrected in the revised manuscript. 

In Figure 4, the panels, their description, and an explanation of what f MLP means are missing.

We thank the reviewer for identifying this mistake. This basophil activation assay kit comes with two types of stimulations. One is the Anti-Fc Epsilon R1 (anti-FcεRI) and the other one is fMLP. We have used FcεRI to stimulate basophils and have mislabeled it as fMLP. Our sincere apologies for this error and we have corrected it in the revised- manuscript.  

In subsection 3.5, what does it mean that the cells tolerated the treatment up to 250 nm? The concentration of the compound assuring cell viability >70% justifies the use of such a concentration in the in vitro studies.

Prior to use the IPX-18 in RBL-2H3 cells we tested the cell viability of these cells in presence of different doses of IPX-18 for 24 h. We accept the suggestion and have revised the statement. We have used a safe level of up to 250 nM where the viability of the cells was unaltered by both IPX-18 and DNP-BSA stimulators.

Moreover, we have also included Annexin assay data to check if any early apoptosis was induced by the compound (Supplementary figure 3) as suggested by the other reviewer.

In subsection 3.7. what does the p50 bound mean? How is this binding region related to the described by authors apoptosis and inflammation? Does the 'small molecule' term always refer to the IPX-18 compound? It should be clear that the authors describe this compound. The description should also highlight how binding IPX-18 to the p50 subunit could theoretically affect inflammatory processes and why this particular subunit was studied and not p65.

P50 bound here refers to the crystal structure we retrieved from the PDB databank. Since our retrieved pdb structure contains a DNA molecule, we used the term “bound”.  The reason for using this structure is to target the same region, i.e., the DNA binding region of p50, as we have shown in previous publication to induce apoptosis. Moreover, Driessler F, et al. have shown that inhibiting p50 nuclear translocation and its subsequent DNA binding is anti-inflammatory. We have now included this statement in the discussion section of the revised manuscript. REF: PMID: 14678266- New ref number 29 in revised manuscript.

In subsection 3.9. the authors present the phosphorylation of the p65 subunit. This subunit of NF-kB dimer is being phosphorylated directly in IKK-independent (atypical) signaling (after UV light, for example), while canonical NF-kB signaling is TNF alpha-mediated. The authors should also note whether and how Nrf2 activation is related to NF-kB and whether Nrf2 activation is beneficial in cancer cells.

It is a very intriguing question. It has been previously shown that NF-kB/p65 antagonizes Nrf2-ARE signaling (PMID: 18241676). Nrf2 activity is critical for the body’s defense against oxidants, carcinogens, and inflammatory insults, and over expression of p65 suppresses Nrf2 levels/activity. Thus, inhibiting p65 activity upregulated Nrf2 levels, and in this way exhibits a protective effect. As suggested, we have included this in the discussion part. (New reference number 40 in the revised manuscript).  

Are Nrf2 results from whole cell extracts? Description of Figure 3.9. suggests a statistical significance that is not indicated in the figure.In general, all figures' captions should be more detailed.

Whole cells were used for flow cytometric analysis of both these proteins as described in sections 2,2,14. We have now indicated statistical significance in the figure, and have elaborated legend. 

What justifies the use of different concentrations of the compound in various presented methods?

We have used log concentrations of IPX-18 in all methods wherever IC50 was determined. We used increasing concentrations of IPX-18 in RBL-2H3 cells to test cell viability. As no change in RBL-2H3 cell viability was noted up to 250 nM concentration, the dose dependence activity of IPX-18 was tested within this range. For flow cytometry -NFkB and NRF2 assays, we tested the mid concentration (100 nM) to verify the bioinformatics observations in invitro. We have included this statement in the revised manuscript so that it would justify the dose selection in different methods. 

Round 2

Reviewer 1 Report

Nil

Author Response

We thank the reviewer for the valuable comments 

Reviewer 2 Report

The authors introduced a majority of requested remarks, although, in this version, the work still needs some changes to be introduced before the manuscript gets published in the Biomedicines

I have included my comments on the manuscript in the comments to authors section and below to the Editor:

  1. A simple summary (lines 10-11) still states that corticosteroids are the main-line anti-inflammatory drugs. Also, the sentence in lines 13-14 is not clear.
  2. Abstract's background in the first sentence states "inflammatory drugs" while the correct name is anti-inflammatory.
  3. Line 40: Brain damage is not a disease, it is an injury that causes the destruction or deterioration of brain cells. Additionally, the listed conditions are not the main inflammation-mediated diseases. I suggest the authors revise this part of their introduction as indicated previously.
  4. The introduction needs to be improved on the Nrf2 role in inflammatory responses.
  5. Lines 49-52: There is a significant mistake, namely, the stimulation with either IL-1β or TNFα activates the IκB kinase (IKK) complex which then mediates phosphorylation, ubiquitination, and degradation of the IκB molecule, then NF-κB might be released and translocated to the nucleus, where it binds to κB motifs in the promoters and induces transcription of target genes. Not the other way around.
  6. Some interactions between Nrf2 and NK-kB signaling pathways are already confirmed (e.g. GSK3 beta) and need to be mentioned.
  7. According to my previous revision, the introduction needs to indicate if the compounds from the group of arylidene indanones already have a proven anti-inflammatory effect, and if so, what kind.
  8. 3.2. subchapter and supplementary figure caption need to be corrected because of grammatical errors.

Author Response

Response: We thank the Academic Editor, the editorial office and reviewers for accepting our revision 1. We welcome this report and have amended all the minor corrections as suggested. The response for each comment is addressed followingly.

1. A simple summary (lines 10-11) still states that corticosteroids are the main-line anti-inflammatory drugs. Also, the sentence in lines 13-14 is not clear.

Response: Accepted and have revised these sentences.

  1. Abstract's background in the first sentence states "inflammatory drugs" while the correct name is anti-inflammatory.

Response: Accepted and corrected.

  1. Line 40: Brain damage is not a disease, it is an injury that causes the destruction or deterioration of brain cells. Additionally, the listed conditions are not the main inflammation-mediated diseases. I suggest the authors revise this part of their introduction as indicated previously.

Response: Comment is welcome and we have revised this part as suggested.

  1. The introduction needs to be improved on the Nrf2 role in inflammatory responses.

Response: Included as advised with new references included.(New references 6-13)

  1. Lines 49-52: There is a significant mistake, namely, the stimulation with either IL-1β or TNFα activates the IκB kinase (IKK) complex which then mediates phosphorylation, ubiquitination, and degradation of the IκB molecule, then NF-κB might be released and translocated to the nucleus, where it binds to κB motifs in the promoters and induces transcription of target genes. Not the other way around.

Response: Thanks for indicating this. We have now modified the statement.

6. Some interactions between Nrf2 and NK-kB signaling pathways are already confirmed (e.g. GSK3 beta) and need to be mentioned.

Response: We welcome the suggestion and have included it in the introduction part with several new references. (New references 6-13)

  1. According to my previous revision, the introduction needs to indicate if the compounds from the group of arylidene indanones already have a proven anti-inflammatory effect, and if so, what kind.

Response: We have included this as suggested. (Ref number 19). 

  1. 3.2. subchapter and supplementary figure caption need to be corrected because of grammatical errors.

Response: Corrected as recommended.
